



# Triple frequency radar retrieval of microphysical properties of snow

Kamil Mroz[1], Alessandro Battaglia[2,1], Cuong Nguyen[3], Andrew Heymsfield[4], Alain Protat[5], and
Mengistu Wolde[3]

[1]National Centre for Earth Observation, University of Leicester, Leicester, United Kingdom
[2]Department of Environment, Land and Infrastructure Engineering, Politecnico di Torino, Turin, Italy
[3]Flight Research Laboratory, National Research Council Canada, Ottawa, Canada
[4]National Center for Atmospheric Research, Boulder, Colorado
[5]Australian Bureau of Meteorology, Melbourne, Victoria, Australia

**Correspondence:** Kamil Mroz (kamil.mroz@le.ac.uk)

**Abstract.** An algorithm based on triple-frequency (X, Ka, W) radar measurements that retrieves the size, water content and degree of riming of ice clouds is presented. This study exploits the potential of multi-frequency radar measurements to provide information on bulk snow density that should underpin better estimates of the snow characteristic size and content within the radar volume. The algorithm is based on Bayes' rule with riming parameterized by the "fill-in" model. The radar reflectivities

are simulated with a range of scattering models corresponding to realistic snowflake shapes. The algorithm is tested on multi-frequency radar data collected during the ESA-funded Radar Snow Experiment. During this campaign in-situ microphysical probes were mounted on the same airplane as the radars. This nearly perfectly collocated dataset of the remote and in-situ measurements gives an opportunity to derive a combined multi-instrument estimate of snow microphysical properties that is used for a rigorous validation of the radar retrieval. Results suggest that the triple-frequency retrieval performs well in estimat-

ing ice water content and mean-mass-weighted diameters obtaining root-mean-square-error of 0.13 and 0.15, respectively for $\log_{10} IWC$ and $\log_{10} D_m$. The retrieval of the degree of riming is more challenging and only the algorithm that uses Doppler information obtains results that are highly correlated with the in-situ data.

## 1   Introduction

Quantifying snowfall rates is essential for understanding the water cycle in mid and high altitudes. Solid phase precipitation

affects many aspects of human life. On one hand, it can represents a hazard to several public services (e.g.: transport, energy distribution networks) as well as private properties; on the other hand, snow accumulations and its eventual runoff is important for hydroelectric power generation and water resource management (Skofronick-Jackson et al., 2019). Snow cover plays a very important role in the climate system modifying the global and regional energy budget due to its high scattering albedo. Despite the undeniable importance of precipitation in the solid phase, there is large discrepancy between different snowfall

accumulation estimates (Mroz et al., 2021b) which reflects a high degree of uncertainty of these products.

   To reduce the uncertainties related to the snow modeling, observational data are needed but these are still rare due to their cost and the remoteness of high-latitude regions where most of the snowfall occurs. Moreover, in-situ measurements at the ground are affected by problems like under-catch, wind-blown snow biases (Fassnacht, 2004) and they are only representative



of the environment around the data collection site. Radar measurements offer better spatial and temporal coverage but their

interpretation is subject to errors/uncertainties that follow from the assumptions made about the scattering properties of the targets in the radar volume and those depend on the snow particles size, density, shape and structure (e.g. Kuo et al. (2016); Eriksson et al. (2018)).

Because different frequency radars respond differently to the microphysical properties of snow (once their wavelengths become comparable with the size of snow aggregates) multi-frequency algorithms were recognised as a potential tool for solid

phase precipitation studies (Hogan et al., 2000; Kneifel et al., 2011). Over the years, the availability of data from complex multi-frequency Doppler radar systems has fostered the development of  algorithms based on dual frequency reflectivity (e.g., Matrosov, 1998), triple frequency reflectivity (e.g., Leinonen et al., 2018; Tridon et al., 2019; Battaglia et al., 2020b), dual frequency reflectivity and Doppler measurements (e.g., Mason et al., 2018) and even full Doppler spectral information (e.g., Mroz et al., 2021a). An increase in number of observables included in the inversion schemes went hand in hand with an

increase in number of retrieved microphysical parameters. For instance, the most recent algorithms aim at quantifying the ice water content, the characteristic size and the bulk density of snow in the radar volume.

This study presents an algorithm for estimating the following microphysical snow properties: mean-mass-weighted diameter, ice water content and degree of riming. The retrieval utilises triple-frequency radar measurements and is based on Bayes' rule. It does not assume any functional form of the particle size distribution but it is based on several datasets collected during

historical airborne campaigns. More detail on the methodology can be found in Sect. 2. Validation of the retrieval, with nearly perfectly collocated in-situ and remote sensing measurements, is presented in Sect. 3. Additionally, we compare the results for different combinations of radar observables. Conclusions are drawn in Sect. 4.

## 2   Methodology

### 2.1   Theoretical basis

The equivalent reflectivity factor for a radar operating at the wavelength $\lambda$ is given by:

$$Z_e = \frac{\lambda^2}{\pi^5 |K_w|^2} \int \sigma_b(D) N(D) \, \mathrm{d}D, \tag{1}$$

where $\sigma_b$ is the backscattering cross section of the particle, $D$ is its diameter, $N$ is the particle size distribution and $K_w$ is the dielectric factor of liquid water at a reference temperature and frequency. In this study, it is assumed that $|K_w|^2 = 0.93$ which is a good approximation for standard temperatures and frequencies below the Ka-band. The reflectivity is usually expressed in

$\mathrm{mm}^6\mathrm{m}^{-3}$ or, due to its high variability, in logarithmic units of $\mathrm{dBZ} = 10\log_{10}(\mathrm{mm}^6\mathrm{m}^{-3})$.

The radar signal, regardless of its frequency, is attenuated along the beam propagation path. This effect is ignored in this study, due to the relatively short path through the ice cloud and thus negligible extinction even at the most affected W-band (Protat et al., 2019). In fact, the distance to the nearest range gate from which the data are collected, is so short the attenuation correction would be smaller than the uncertainty of the measurements themselves. In case of longer distances attenuation

corrections must be performed (e.g. Kalogeras and Battaglia (2021)) before our algorithm can be applied.





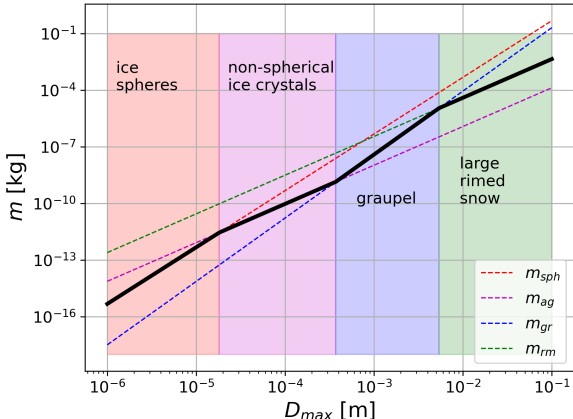

**Figure 1.** Mass–size parametrization of Morrison and Grabowski for rimed aggregates. The different colors correspond to the size ranges where a specific power law formula is used: red for small solid ice spheres, magenta for non-spherical ice crystals, blue for fully rimed snow, green for aggregates that are not completely filled with rime.

Due to a large variety of particle size distributions (PSDs) found in nature no explicit analytical formula that approximate their shape is used. Instead, the radar simulations used in this study are based on the binned PSD measurements collected by optical instruments during different in-situ airborne campaigns. In addition to radar reflectivity simulations, selected microphysical properties are prescribed to the each PSD in the dataset. These properties include the ice water content (IWC), the mean mass-weighted snow diameter ($D_m$) and the degree of riming ($\alpha$), defined below. The size of snowflakes is defined in terms of the diameter of the smallest circumscribing sphere. The database constructed in this way defines a statistical mapping between microphysical properties of ice PSDs $[D_m, IWC, \alpha]$ and radar relectivities at the frequencies of interest.

The mass of the snowflakes is modeled using the parametrization of Morrison and Grabowski (2008). In this scheme, riming is modeled by "filling-in" the interstices between ice crystal branches by super-cooled liquid droplets (Heymsfield, 1982). The mass of unrimed aggregates follows the power-law relationship:

$$m_{ag}(D) = \alpha_{ag} D^{\beta_{ag}} \tag{2}$$

where $\alpha_{ag}$ and $\beta_{ag}$ are the parameters of the fit and the physical quantities are in SI units. We assume $\alpha_{ag} = 0.015$ and $\beta_{ag} = 2.05$ which agrees well with in-situ observations (Leroy et al., 2016) and the simulations of aggregates (Leinonen and Szyrmer, 2015). For sizes where the power-law formula would exceed the mass of solid ice spheres the latter is used.

The mass parametrization for particles that underwent a riming process is more complex. The range of ice sizes is divided into four domains: small ice spheres, dense non-spherical ice crystals, graupel (fully rimed aggregates) and large partially rimed snowflakes as it is shown in Fig. 1. The first two groups are the same as those described for unrimed snow flakes, but their specific relationship between mass and size is only applicable up to a certain size. The transition between dense non-spherical ice crystals and graupel occurs at the size where their masses are equal. For diameters exceeding this critical point the mass of





particle is given by:

$$m_{gr}(D) = \alpha_{gr} D^{\beta_{gr}} \tag{3}$$

where $\alpha_{gr}$ and $\beta_{gr}$ are the m-D parameters specific for graupel ($\alpha_{gr} = 469$ and $\beta_{gr} = 3.36$; Leinonen and Szyrmer, 2015). As
a consequence of the "filling-in" conceptual model, this relation applies to particles that are small enough to be fully filled with
rime. For larger sizes, the exponent ($\beta$) of the mass-size relation remains the same as for unrimed aggregates ($\beta_{ag} = 2.05$) and
only the prefactor ($\alpha$) increases i.e.:

$$m_{rm}(D) = \alpha_{rm} D^{\beta_{sn}}. \tag{4}$$

Again, the changeover between graupel and partially rimed aggregates occurs where their masses become equal, which provides
a continuous transition in the $m - D$ formula. The larger the $\alpha_{rm}$ the larger the mass of rimed particles and the size where the
transformation occurs. With this approach, the density if ice particles is completely determined by $\alpha_{rm}$ and this parameter is
used to describe the degree of riming.

## 2.2   Scattering Model

To simulate radar reflectivity the backscattering cross-section of all the particles within the radar volume must be known (see
eq. 1). This is straightforward for rain drops since their shape can be precisely simulated (Ekelund et al., 2020), but is more
complicated for snow due to the variety of snowflake types, sizes, possible arrangements of the ice crystals within a single
aggregate and is some cases a certain degree of riming (Kneifel et al., 2020). For wavelengths much larger than size of snow
"soft" sphere approximation provides a good approximation of the scattering properties (Kuo et al., 2016). However, when
a diameter of a snowflake is comparable to or larger than the wavelength the scattering calculations need to account for all
the aforementioned properties of the ice particle (Tyynela et al., 2011). In this study, we use a combination of three publicly
available scattering data sets: the ARTS database of Eriksson et al. (2018), the OpenSSP database of Kuo et al. (2016) and the
rimed snowflakes simulations of Leinonen and Szyrmer (2015). The fist two data sets cover a variety of snowflake types and
sizes populating our database up to 13 mm in diameter. The data set of Leinonen and Szyrmer (2015) complements the other
two by covering much larger range of snow densities and larger sizes ($D < 25$ mm).

     Despite the fact that the formation of certain types of snow is determined by the atmospheric conditions (REF), ice particles
observed by the radar could be transported there, which makes it very difficult to reliably simulate measurements tailored to
a given snow type. Therefore, we treat all the snowflake types as equally possible and the backscattering properties are given
as a function of their mass and size only, i.e., $(m, D) \mapsto \sigma_b$. The function $\sigma_b$ is constructed by grouping all the snow particles
from the scattering data sets by their mass and size using logarithmically spaced bins along two dimensions. Then, in each
bin, a mean mass-squared-normalised backscattering cross-section is computed by averaging $\sigma_b(D_i, m_i)/m_i^2$ of individual
particles in the bin. Because in the Rayleigh scattering regime $\sigma_b$ is proportional to $m^2$ (Hogan et al., 2012), the introduced
normalisation reduces variability of the averaged variable within the bin and it prevents from biases toward large masses that





would contribute the most to the mean otherwise. The final estimate of $\sigma_b$ in each bin is given by:

$$\sigma_b(m, D) = m^2 \times \underset{i \in bin}{\mathrm{mean}} \left( \frac{\sigma_b(m_i, D_i)}{m_i^2} \right). \tag{5}$$

For an arbitrary value of $m$ and $D$ one needs to interpolate between the mean normalised backscattering values and multiply by mass squared. For sampling points outside the convex hull defined by the range of sizes and masses within our dataset the 110 "soft" sphere approximation is used.

Based on this approach, to simulate the radar reflectivity at a given wavelength, $\lambda$, and for an arbitrary particle size distribution, $N(D)$, it is sufficient to determine the degree of snow riming ($\alpha_{rm}$ in the previous section). Once $\alpha_{rm}$ is set, the relation between the snowflake mass and size is unambiguously determined which allows calculations of the backscattering cross-section area and the equivalent radar reflectivity value follows from eq. (1).

## 115 2.3 Inversion Scheme

The previous section described how to simulate radar reflectivity for a given PSD and a degree of riming. This section focuses on the inverse problem, i.e., given a set of radar measurements how to estimate the properties of the observed PSD. This study focuses on the triple frequency observations that, due to limited information content, prevent a size resolved retrieval of the PSD. Therefore, the inversion scheme presented here aims at estimating only bulk properties of snow in the radar volume. 120 These properties include: a mean mass-weighted diameter ($D_m$), ice water content ($IWC$) and a degree of riming ($\alpha_{rm}$). All of these parameters are positive thus for practical reasons it is more convenient to retrieve their logarithms, thus the state vector is given by:

$$x = [\log_{10} D_m, \log_{10} IWC, \log_{10} \alpha_{rm}]^T. \tag{6}$$

The observation vector is formed from the reflectivity at the smallest frequency and the dual frequency ratios (DWRs), i.e. 125 the differences between reflectivities at different bands in $\mathrm{dBZ}$:

$$y = [Z_X, DWR_{X-Ka}, DWR_{Ka-W}]^T \tag{7}$$

There are several advantages of exploiting the DWR-DWR space. First, the DWRs are independent of the IWC (see eq. 1) of the observed PSD because its dependence cancels out once the difference (quotient in the linear units) of the different bands is taken. This simplifies the inversion scheme because it introduces some degree of orthogonality in the observation space. 130 Secondly, previous studies (Kneifel et al., 2011, 2015) have shown that the DWR-DWR data have a potential to discriminate between different degrees of riming on top of the sizing capabilities. For the uncertainty of the radar measurements $1\,\mathrm{dB}$ random error at all the frequency bands is assumed. Because the radar measurement errors are uncorrelated the error covariance matrix of the observation vector $y$ is:

$$COV = \begin{bmatrix} 1 & -1 & 0 \\ -1 & 2 & -1 \\ 0 & -1 & 2 \end{bmatrix} \tag{8}$$



The retrieval scheme adopted for this study is based on Bayesian theory and aims at estimating the expected value of the state $x$ for a given measurement $y$, i.e.,:

$$E(x|y) = \int_X x\, p(x|y)\, \mathrm{d}x, \tag{9}$$

where $p(x|y)$ denotes the conditional probability of $x$ subject to $y$ being observed. To estimate $p(x|y)$ a dataset of in-situ PSD measurements is used. This database is constructed from the measurements collected during MC3E (Jensen et al., 2016),

IPHEx (Barros et al., 2016), OLYMPEX (Houze et al., 2017) and HAIC/HIWC (Leroy et al., 2015) field campaigns where approximately 0.25 million PSD measurements were taken in total. The HAIC/HIWC campaign was selected to complement NASA-led campaigns to add high ice water content (IWC) measurements. Unlike the other campaigns where the water content was measured with the Nevzorov probe, the HIWC used an isokinetic evaporator specifically designed for high IWC measurements. The measurements of IWC are used as a complementary information to the PSD data which allows us to estimate the

degree of riming by matching the measured WC with the one simulated from the PSD for different values of $\alpha_{rm}$. This procedure establishes a PSD-specific relationship between the mass and size of the observed snowflakes so that the mass-weighted mean diameter ($D_m$) can be estimated and it is included in the in-situ training dataset. For each PSD, the radar reflectivities at X-, Ka- and W-band are simulated using the scattering model described in section 2.2. Then, for any hypothetical measurement $y$, the probability that the PSD resembles this measurement is computed as:

$$p(x^i|y) = \frac{1}{2\pi\sqrt{det(COV)}} \exp\left[-0.5(\delta y_i)^T COV^{-1}(\delta y_i)\right], \tag{10}$$

where $\delta y^i$ is the difference between the hypothetical and the simulated measurement corresponding to the $i$-th element in the in-situ database, i.e.:

$$\delta y_i \quad = \quad y - y_i = \left[Z_X - Z_X^i, DWR_{X-Ka} - DWR_{X-Ka}^i, DWR_{Ka-W} - DWR_{Ka-W}^i\right]^T. \tag{11}$$

By expressing eq. (9) in a discrete form the expected value of $x$ subject to $y$ being observed is given by:

$$E(x|y) = \frac{\sum_i x^i p(x^i|y)}{\sum_i p(x^i|y)}. \tag{12}$$

Theoretical uncertainty of the retrieval is estimated as a weighted standard deviation of the state vectors for a given measurements, i.e.:

$$Var(x|y) = \frac{\sum_i (x^i)^2 p(x^i|y)}{\sum_i p(x^i|y)} - E(x|y)^2. \tag{13}$$

The retrieval presented here is based purely on the database of in-situ measurements and does not assume any analytical

form of the PSD. Moreover, the radar simulations are based on the scattering properties of realistic snowflakes. An example of this inverse mapping for $Z_X = 20$ dBZ is presented in Fig. 2. The characteristics of the retrieval are in line with the previous studies (Kneifel et al., 2015), e.g. low density snow usually occurs for $DWR_{Ka-W} < 10$ dB and $DWR_{X-Ka} > 4$ dB whereas heavily rimed particles occupy regions with low $DWR_{X-Ka}$ or $DWR_{Ka-W} > 12$ dB. The mean mass-weighted snow diameter tends to increase with the DWR values and the largest sizes are observed for low density aggregates.



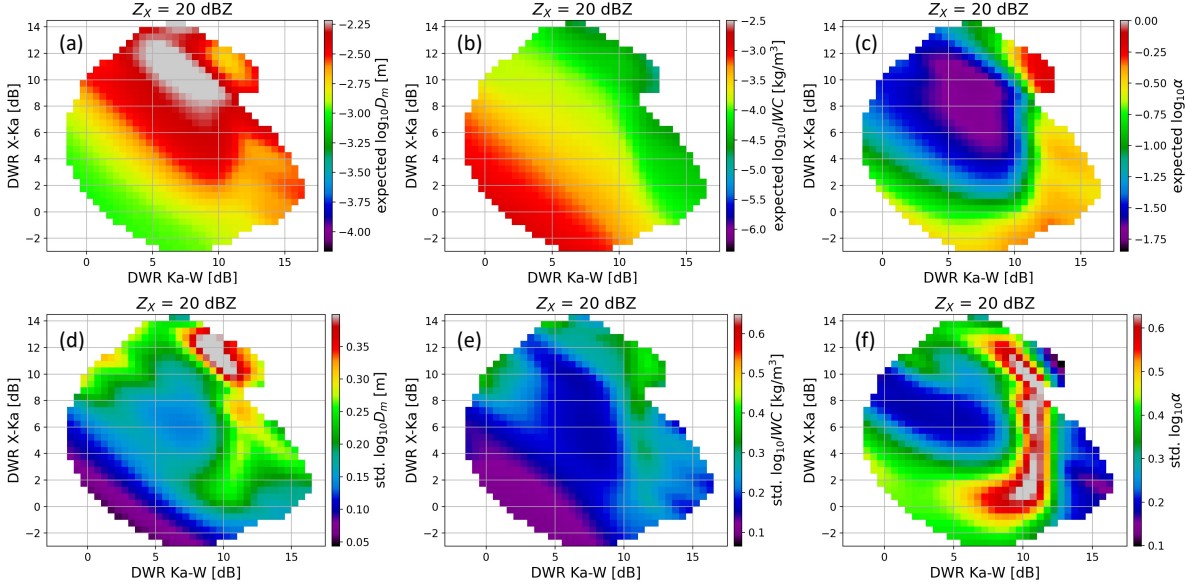

**Figure 2.** The expected values of $\log_{10} D_m$, $\log_{10} IWC$, $\log_{10} \alpha_{rm}$ in the $DWR_{X-Ka}$–$DWR_{Ka-W}$ for $Z_X = 20$ dBZ. This inverse model is derived from the in-situ airborne PSD measurements during MC3E, IPXEx, OLYMPEX and HAIC/HIWC campaigns.

Although, the DWRs do not depend on the IWC it is evident from the plot that, for the same reflectivity, low DWRs correspond to higher ice loads. This shows a compensating effect between $D_m$ and $IWC$: to get the same reflectivity value for small particles the IWC has to be large and vice versa.

The estimates of the uncertainty reveal limited capabilities of the triple-frequency retrievals to accurately quantify bulk ice density except for the signatures of extreme aggregation or strong riming mentioned before. The transition between these two

distinct regimes is characterised by very large uncertainty in $\log_{10} \alpha_{rm}$ that reach 0.65 (more than a factor of 4) for $Z_X = 20$ dBZ. The other two state variables are much better constrained by the measurements; however, an elevated uncertainty is also observed for the transition region between heavy riming and aggregation domains.

The methodology presented in this section can be also applied to an arbitrary set of measurements. In particular, a single frequency retrieval can be constructed and compared to the multi frequency one. We perform this exercise with an X-band

retrieval that is validated in the next section. We also test a retrieval where triple-frequency frequency reflectivity data is supplemented by mean Doppler velocity measurements.





## 3 Application and Validation

### 3.0.1 SnowRadExp dataset

The Radar Snow Experiment For Future Precipitation Mission (RadSnowExp; Wolde et al., 2019) research flights were con-
180 ducted in mid-latitudes and near the Arctic circle (Iqaluit, NU, Canada, 63N), during the fall of 2018, covering a wide geo-
graphical region and microphysical conditions. The flights focused on sampling precipitation systems where large aggregates
and rimed particles were present in order to optimize the triple-frequency analyses. Multi-frequency radar observations were
obtained from nadir and zenith looking antennas of the NRC Airborne W and X-band (NAWX) radars (Wolde and Pazmany,
2005) and the University of Wyoming's Ka-band Precipitation Radar (KPR; Haimov et al., 2018). The NAWX antennas are
185 housed inside an unpressurized blister radome mounted on the right side of the aircraft fuselage and the KPR radar was in-
stalled on the left wingtip pylon. Although the three radars are on the same platform and almost collocated, mismatched radar
beamwidths and differences in vertical resolutions and radar data dwell times required additional processing steps to provide
the best possible matching of the radar volumes to reduce the DWR estimation errors (Nguyen et al., submitted, 2021). In
addition to the radars, the NRC Convair-580 aircraft was equipped with a wide array of state-of-the-art in-situ sensors for mea-
190 surements of aircraft and atmospheric state parameters, and cloud microphysical properties. Bulk liquid water content (LWC)
and total water content (TWC) were measured simultaneously with single-particle size distribution, ranging from small cloud
droplets to large precipitation hydrometeors. In this work, cloud particle size distribution was composed using a combination
of data from several single-particle probes: Fast Cloud Droplet Probe (FCDP, 2-50 µm, SPEC Inc.), two-dimensional stereo
(2DS, 10-1200 µm, SPEC Inc.) probe, and vertically oriented High Volume Precipitation Spectrometer version 3 (HVPS3,
150-19200 µm, SPEC Inc.) probe or Precipitation Imaging Probe (PIP, 100-6400 µm, DMT). TWC and LWC were measured
by the Nevzorov, a constant-temperature hot-wire probe (Korolev et al., 1998). We estimate the accuracy of the Nevzorov data
during RadSnowExp to be on the order of $0.05~\mathrm{gm^{-3}}$.

### 3.0.2 Validation data

The data collected during the SnowRadExp campaign offer an unprecedented opportunity to validate the triple-frequency radar
snow retrieval since the remote measurements are well collocated with the in-situ observations of snow microphysics. Having
said that, a gap between the probes and the first radar range gate of approximately $100~\mathrm{m}$ can introduce some uncertainty in
the validation process due to the vertical gradients in the snow microphysics and therefore in the reflectivity. For some parts of
the validation flight, the difference between the radar return below and above the plane can reach $10~\mathrm{dB}$. In such a situation it
is difficult to decide which measurement resembles best the microphysics at the flight level.
To address the collocation issue, an optimal estimation framework (Rodgers, 2000) is used to provide the most likely estimate
of the state variables. The only variable that is estimated here is a degree of riming ($\log_{10}\alpha_{rm}$) while it is assumed that the
PSD measured by the optical probe is errorless. The a-priori estimate of $\log_{10}\alpha_{rm}$ is based on the in-situ dataset described in
Sect. 2.3, where the mean value of $-1.31$ and the standard deviation of $0.43$ is observed. The measurement vector is composed
of the $\log_{10}IWC$ as it is measured by the Nevzorov probe and the average of the reflectivity above and below the aircraft at X,





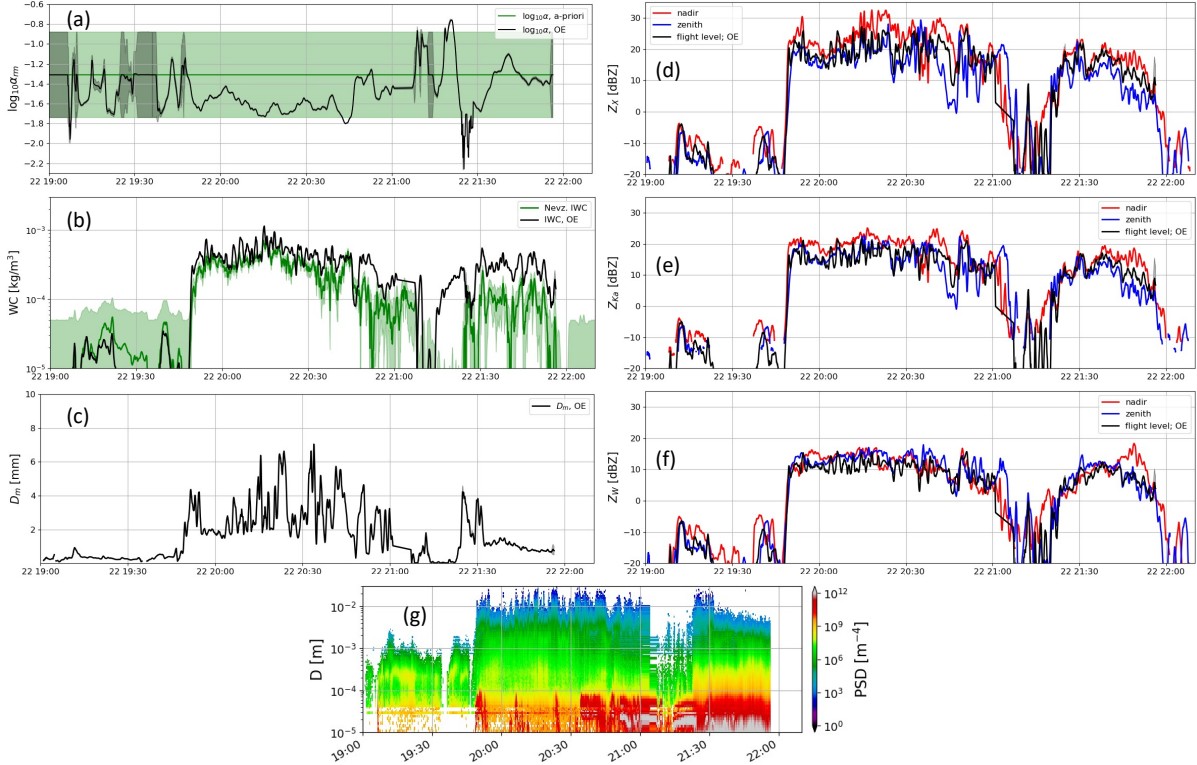

**Figure 3.** The estimates of the microphysical properties of snow and the radar reflectivity at the flight level. Panel (a) shows the retrieved degree of riming in black and its prior estimate in green. The retrieval is based on the data from triple frequency radar, Nevzorov probe and PSD measurements. Panel (b): the estimated IWC (black) and Nevzorov probe data (green). Panel (c): the estimated mean mass-weighted diameter of snow. Panels (d,e,f) show the reflectivity at the X-, Ka- and W-band, respectively. The red (blue) line shows the measurements below (above) the plane while the black line is the estimate at the flight level that is in agreement with the microphysical properties shown on the left and the collected PSD data (panel g). Shading represents the uncertainty of each estimate.

Ka and W bands. The corresponding uncertainties of the measurements are $0.05~\mathrm{g/m^3}$ and one half of the difference between the radar measurements. In case the reflectivity difference between both sides of the plane is within $2~\mathrm{dB}$ we set the uncertainty of the measurement to $1~\mathrm{dB}$ to account for random errors in the radar observation. The forward model to simulate the IWC and the radar reflectivity at the frequencies of interest are those described in the methodology section.

Figure 3a depicts the results of the OE retrieval for a flight leg. The black line with shading represents the retrieved value

of $\log_{10}\alpha_{rm}$ and its estimated standard deviation, respectively. The remaining panels show the other microphysical properties (on the left) and the radar reflectivities (on the right) with their associated uncertainties estimated by propagating the errors on $\alpha_{rm}$. For the first half of the flight (until 20:45UTC) the optimally estimated IWC is in agreement with the Nevzeorov probe data but later it tends to be higher than the in-situ instrument reports. This discrepancy mainly occurs where the measured IWC is relatively low and so the associated uncertainty is large. The estimates of the reflectivity at all the considered frequency

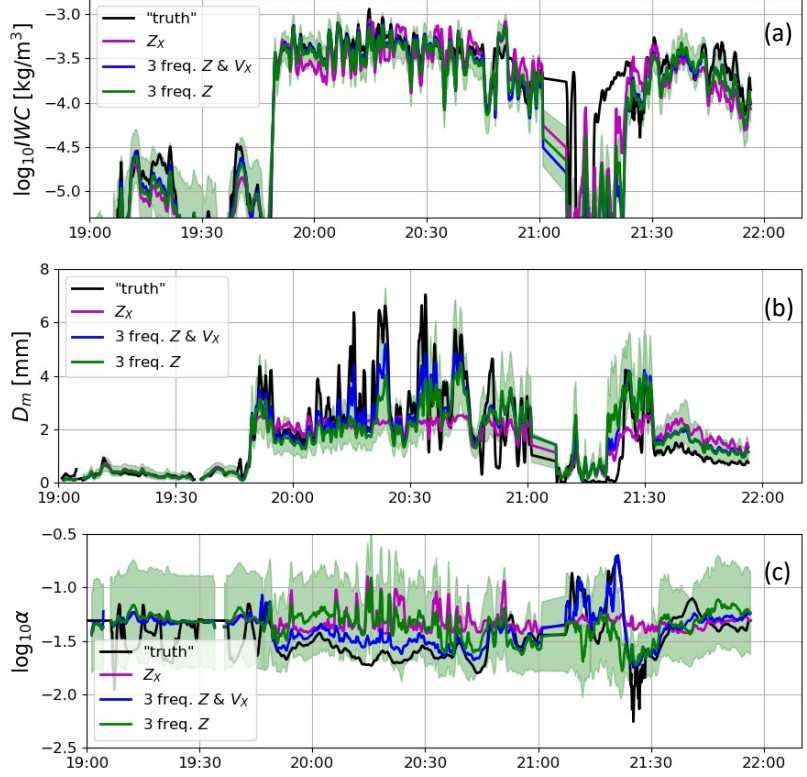

**Figure 4.** The results of different microphysical retrieval along the validation flight. The black line shows the snow microphysical properties estimated derived from the PSD measurements, Nevzorov probe and radar data. The green, blue and magenta lines correspond to the retrievals based on the triple frequency reflectivity, triple frequency reflectivity with the mean Doppler velocity at the X-band and the algorithm based on the X-band reflectivity only, respectively. The green shading shows the uncertainty of the triple-frequency retrieval.

bands are in agreement with the measurements (see blue and red lines in Fig. 3) and the estimate of their uncertainty is within $1\,\mathrm{dB}$ for the most of the flight.

### 3.0.3  Validation of the snow microphysical retrieval

The optimally estimated radar measurements and the microphysical data described in the previous subsection are used for the validation of the microphysical retrieval. First, the triple frequency radar reflectivity at the in-situ flight level are used to form

the measurement vector (see eq. 7). Then, the expected value of the state vector is estimated using the methodology presented in Sect. 2.3. Finally, the retrieval results are evaluated against the microphysical properties determined by the optimal estimation framework at the flight level. An analogous analysis is repeated for two other retrievals: one that is based on single frequency X-band radar reflectivity only and another one based on triple frequency reflectivity data with the addition of the mean Doppler velocity at the X-band.



The results of the three retrievals in comparison to the in-situ "truth" along the flight are shown in Fig. 4. At first glance, all the retrievals perform well in estimating IWC; nevertheless multi-frequency algorithms show some advantages in certain parts of the flight. The retrieval of the mean mass-weighted diameter shows more differences between the algorithms. Clearly, the single-frequency algorithm struggles in retrieving large snow sizes ($D_m > 2.5$ mm). This indicates that the radar reflectivity at the X-band is correlated more with the water content than with the size when large snowflakes are present in the radar

volume. The triple-frequency reflectivity based retrieval performs better in estimating $D_m$ but it is not as close to the truth as the algorithm that uses the Doppler information. This improved size estimation capability is the result of additional information on ice density provided by the velocity of particles.

A more quantitative retrieval evaluation is presented in Fig. 5. The bulk retrieval statistics are displayed in the top left corner of each panel. These statistics were produced for the validation points where $Z_X > -20$ dBZ, $DWR_{X-Ka} > 1$ dB and

$DWR_{Ka-W} > 1$ dB. These conditions were imposed to show the difference between the triple-frequency algorithm and the single-frequency one. For negligible DWRs, multi-frequency information is reduced so the difference between the algorithms. As an additional constraint we require the IWC is at least 70% of the total water content, which removes points where $D_m$ estimates are negatively biased by an abundant number of small liquid drops in the PSD data.

Out of all the microphysical parameters the one that is the most difficult to retrieve is the degree of riming. The single-

frequency retrieval is uncorrelated with the in-situ data which indicates the minimal information content on the snow density of these measurements, at least in the range of the observed reflectivity values ($-20 < Z_X < 30$ dBZ). The triple-frequency radar reflectivity observations contain more information about the density of ice. This is reflected in a positive but still low Pearson correlation coefficient of 0.28. The triple-frequency radar measurements are not enough to constrain the inverse model enough to estimate this parameter with high accuracy. This has been already suggested by the uncertainty estimates of this

parameter presented in Fig. 2f. The DWR-DWR space separates well the extreme signatures of riming and aggregation but the intermediate values of $\log_{10} \alpha_{rm}$ cluster around $DWR_{Ka-W}$ of 10 dB which results in high uncertainty of this parameter. These results confirm the findings of Mason et al. (2019) who showed that it is very challenging to disentangle the effects associated with changes of the PSD and of the snow density just looking at DWR-DWR plots. Therefore, we recommend caution when interpreting DWR-DWR data. The best retrieval of the degree of riming is achieved when a reliable mean

Doppler velocity estimates is available. In this case, the correlation between the truth and the retrieved values increases to 0.85 and the root-mean-square-error (RMSE) drops by a factor of 3 compared to the reflectivity only based algorithm.

The accuracy ranking of the retrievals of $D_m$ and $IWC$ is identical to that for $\alpha_{rm}$. The single-frequency retrieval gets the lowest score, the second place goes to the triple-frequency one and the triple-frequency Doppler algorithm performs the best. All the retrievals of $D_m$ are strongly correlated with the in-situ data. Clearly, the X-band reflectivity based algorithm

underestimates the highest end of the snow sizes. The triple-frequency one also tends to underestimate the large sizes but the underestimate is smaller as it is shown by the reduced RMSE value. The underestimate is not as systematic as for the single-frequency retrieval but the validation data is more scattered for large values which reflects uncertainties of the algorithm. There is an additional improvement in the accuracy of the $D_m$ retrieval if Doppler measurements are included. These observations facilitate the estimation of the ice density and thus reduce the uncertainty in the characteristic size of snow. The bi-modal



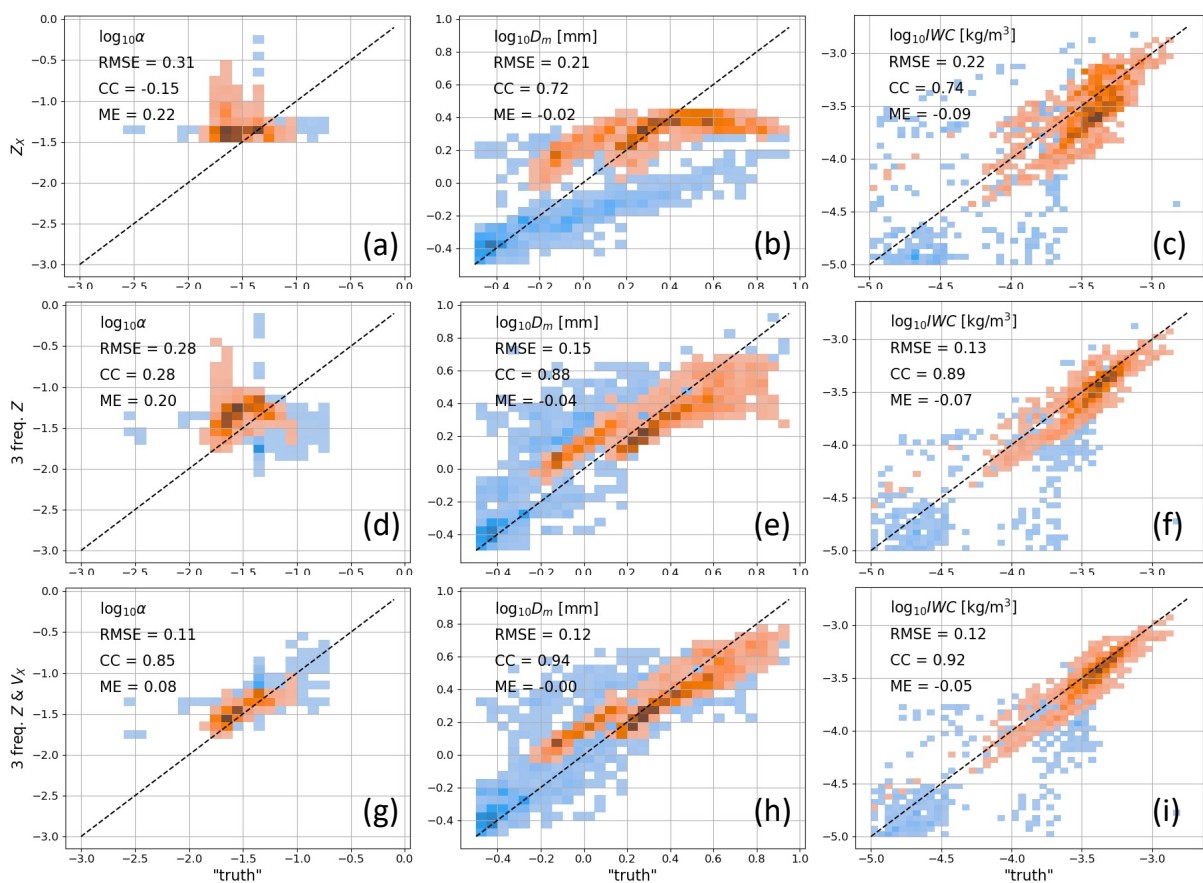

**Figure 5.** Histograms of in-situ measurements (x-axis) collected during the RadSnowExp campaign and the retrieved (y-axis) microphysical parameters. Different columns correspond to different microphysical properties, i.e. $\log_{10} \alpha_{rm}$, $\log_{10} D_m$, $\log_{10} IWC$. The rows correspond to different combination of radar measurements: single frequency reflectivity; triple frequency reflectivity and triple frequency reflectivity with Doppler data. The blue pixels represent the data along the whole flight while the red ones represent measurements where $Z_X > -20$ dBZ, $DWR_{X-Ka} > 1$ dB, $DWR_{Ka-W} > 1$ dB and the IWC is at least 70% of the total water content. The bulk statistics shown in the top left corner of each panel correspond to the red pixels.





clustering of the validation points for $D_m$ results from overestimating the size of small snowflakes ($\log_{10} D_m < 0.2$) by all
the presented algorithms. These sizes correspond to PSDs measured after 21:30UTC and they are characterised by a very high
concentration of particles smaller than 0.1 mm. Such small ice crystals do not generate any DWR signal but their concentration
is large enough to produce $D_m$ smaller than expected from the global statistics. This pinpoints at the shortcomings of the multi-
frequency radar based approaches, i.e., their sizing capabilities are limited to the parts of the PSD where at least one of the
frequency bands is in non-Rayleigh regime. An unusual concentration of small ice crystals for a given $D_m$ can affect the
accuracy of the estimate. The retrievals of IWC show very similar bulk statistics as the one for $D_m$. None of the algorithms
seem to be affected by a systematic bias with larger uncertainties for the single-frequency algorithm.

## 4 Conclusions

A methodology to estimate some important bulk microphysical properties of snow is presented and evaluated using in-situ
airborne data. The retrieval algorithm is based on the Bayes theorem, where the expected values of the microphysical properties
for a given set of the radar measurements are estimated from a dataset of airborne in-situ flights and corresponding radar
reflectivity simulations. In this study, we focus on the triple-frequency reflectivity retrieval. The capabilities of the algorithm are
tested with the data collected during SnowRadExp campaign in Canada. Advantages and limitations of the retrieval are shown
by contrasting the performance of the algorithm with two possible alternatives: the algorithm based on X-band reflectivity only
and the retrieval where triple-frequency reflectivity data are complemented by the mean Doppler velocity information.

The evaluation results indicate that the single frequency (X-band only) algorithm can be effectively used to estimate the
ice water content in the radar volume but, not unexpectedly, with uncertainties larger than for multi-frequency approaches.
The estimate of the mean-mass-weighted diameter saturates at around 3 mm which results in a negative bias for larger sizes.
This demonstrates that the size of snow and the radar reflectivity are not well correlated in presence of large snowflakes. In
stratiform precipitation conditions, that were sampled during the flight used for validation, single frequency radar measurement
do not constrain the retrieval of the ice density which tend to oscillate around the global mean statistics.

The main advantage of triple-frequency approaches over single-frequency is their capability to better represent the in-situ
estimates of the mean-mass-weighted diameter especially for the characteristic sizes greater than 3 mm. Moreover, they are
characterised by higher accuracy of the IWC estimates (0.22 vs 0.13 root-mean-square-error of $\log_{10} IWC$). In contrast to the
single-reflectivity algorithm, the multi-frequency one presents some skills in retrieving the degree of riming of the observed
snow. However these abilities are limited to only cases with extreme aggregation or riming. Intermediate regimes are very
difficult to distinguish from the signatures in the DWR-DWR space that can be produced either by varying the shape of the
particle size distribution or the ice density. This ambiguity results in high uncertainty in the estimates of the degree of riming
and low correlation coefficient (0.3) between the in-situ data and the retrieval.

The triple-frequency reflectivity retrieval that also ingests Doppler information performs the best out of all the analysed
algorithms. In stratiform conditions, it retrieves accurately the degree of riming reaching a root-mean-square-error of $\log_{10} \alpha_{rm}$



of 0.11. The retrieved degree of riming is strongly correlated with the validation data (CC=0.85). This capability helps in further improvements of the estimates of the size and water content of the observed snow PSD.

The analysis presented here takes advantage of a validation dataset that is estimated via optimally matching the in-situ measurements of the water content measured by the Nevzorov probe, the PSD measurements collected by optical array probes and the remote sensing data from triple frequency radars. This unique dataset provides an unprecedented opportunity to validate multi-frequency radar retrievals of the snow microphysics. The application of the methodology is restricted to one flight only and should be applied to long-term observations in order to produce more statistically significant results. Future studies could also consider the inclusion of radars in the G-band (Lamer et al., 2021) and assess their impact on the retrieval of smaller particles.

Note that, the radar observations at the flight level used for the retrieval were not directly measured by the radar. They were simulated using the same forward model as for the in-situ PSD dataset. This has two consequences. First, this approach is equivalent of assuming that the forward model is error free, i.e., the scattering properties of snow depend only on its size and mass and snow riming can be parametrised with one continuous parameter only. Secondly, the vector of observables used for validation is also free of the random errors that affect real measurements. In particular, the Doppler measurements are assumed to be unaffected by the vertical air motion that can substantially alter real data. Although this approach results in the error estimates presented here being underestimated, it shows capabilities and limitations of different radar setups when no assumption on the PSD shape is made which is the goal of this paper.

A key finding of the study is that the Doppler capability is essential to estimate the density of snow in the radar volume which remains the biggest challenge in the accurate quantification of the ice phase precipitation. Based on these findings, we strongly recommend considering Doppler capabilities for future space-borne radar missions (Battaglia et al., 2020a) aimed at characterizing solid phase precipitation.

*Data availability.* The OpenSSP and ARTS datasets are publicly available at https://storm.pps.eosdis.nasa.gov/storm/OpenSSP.jsp and https://doi.org/10.5281/zenodo.4646605, respectively. The backscattering cross-sections of rimed particles are stored in the supplementary data Table S1 of (Leinonen and Szyrmer, 2015). The cloud microphysics data collected during OLYMPEx, IPHEx and MC3E campaighns are available online from the NASA Global Hydrology Resource Center DAAC, Huntsville, Alabama, U.S.A.: http://dx.doi.org/10.5067/GPMGV/OLYMPEX/MULTIPLE/DATA201, http://dx.doi.org/10.5067/GPMGV/IPHEX/MULTIPLE/DATA201, http://dx.doi.org/10.5067/GPMGV/MC3E/MULTIPLE/DATA201. The data collected during HAIC-HIWC are available upon request at https://data.eol.ucar.edu/master_lists/generated/haic-hiwc.

*Author contributions.* KM led the preparation of the paper, developed the retrieval scheme and performed the analysis presented here. AB advised on the methodology of the algorithm. CN and MW processed and provided the SnowRadExp validation dataset. AP and AH analysed and processed the PSD citation data collected during different campaigns.



*Competing interests.* The authors declare that they have no conflict of interest.

*Acknowledgements.* The work by Kamil Mroz was performed at the University of Leicester under grant no. RP1890005 with the National Centre for Earth Observation. The work done by A. Battaglia was funded by the US Atmospheric System Research (grant no. DE-SC0017967). C. Nguyen and W. Mengistu were funded by the ESA-project "Raincast" contract: 4000125959/18/NL/NA. The essential contributions of Water Strapp (Met Analytics Inc.), Thomas Ratvasky (NASA), Lyle Lilie (Science Engineering Associates), Craig Davison (National Research Council Canada), Fabien Dezitter (Airbus), and Delphine Leroy (Laboiratoire de Meteorologie Physique) to the production of the HAIC-HIWC Darwin dataset are warmly acknowledged.

This research used the ALICE High Performance Computing Facility at the University of Leicester.





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
