# Peer review of "Triple frequency radar retrieval of microphysical properties of snow"

_Atmospheric Measurement Techniques, 2021_

## Author Comment (AC1)

This manuscript describes an approach for triple frequency radar retrievals of precipitating ice microphysical parameters. It is overall an interesting study worth of publishing after the authors address comments below.

Comments.

1. Equation (2): m-D relations. Usually, size of particles is defined in terms their major dimension. Since you define it here differently, it would be useful if you briefly discuss what typical differences can be expected between your size definition and that which uses the major dimension. You are suggesting on line 68 that your relation agrees with the one from Leinonen and Szyrmer (2015) but these authors (unlike you) use maximum size for D, so discussion about differences in size definition would be helpful. Also, it appears that your relation m=0.015D^2.05 [SI units] provides particle mass values, which are quite a bit different than many existing relations for aggregates (see for example, Mitchell JAS 1996, p.1716, relations from Heymsfield et al. JAS 2010, p.3303 and many others). Again, Fig. 1 shows mass dependence on Dmax not the size you use according the statement on line 61. Can you address these issues?

We state explicitly what we understand by the maximum diameter because the definition is ambiguous and it differs between the studies, e.g., McFarquhar and Heymsfield (1996) defined the Dmax as the largest particle dimension along the main flow; Leroy et al. (2016) defined Dmax as the largest length through the center of the particle image. The definition of particle size used in our article coincides with this one used by Leinonen and Szyrmer (2015) as it is written in the last sentence of their Section 2.4. The same definition is also used by Heymsfield et al. (JAS 2010) in a two-dimensional setting for analysis of images of ice crystals. Because the scattering simulations and the in-situ PSD datasets both use this definition we adopted it for our study. Regarding the mass-size formula it must be noted that it is only the beta parameters that is constant throughout the manuscript and its value of 2.05 is in agreement with commonly used values in literature where it ranges from 1.8 to 2.2. The value of 0.015 for alpha is just a baseline. If smaller values of alpha are retrieved this is interpreted as less dense particles; if alpha is larger more dense/rimed snowflakes are expected in the radar volume.

2. Do you account for particle orientations and shapes? Observations show that DWR depends quite strongly on particle shape and orientation (e.g., Matrosov et al. JAMC 2019 p. 2005). For vertical beam measurements, more spherical particles produce larger DWR than less spherical particles.

We do not account for particle orientations and shapes. Two of the scattering models we use (ARTS and OpenSSP) assume random orientation of particles whereas in the dataset of Leinonen and Szyrmer (2015) snowflakes are horizontally aligned along the major-axis dimension. Despite this inconsistency these datasets are combined to form a scattering table that covers a wide range of snowflake sizes and masses. The expected scattering properties of a snowflake for a given mass and size are computed as an average of scattering properties of particles from the database in the neighbourhood of that point, regardless of particle shape or orientation. This approach provides an approximation for an

ensemble of snowflakes which mimics somewhat radar measurements of large air volumes where different ice habits can mix together.

3. I think the NRC aircraft can also provide measurements with side view radar beams. Did you compare side and vertical measurements?

The NRC W-band and X-band radars have side-looking antennas but the triple frequency radar data are only available in nadir and zenith directions therefore we didn't use side view measurements in the analysis.

4. From what I know, the NRC aircraft microphysical suite has two Nevzorov probes (at least it was the case with the flights I know about). The IWC estimates form the two probes can differ. Did the flights, which you analyzed, have measurement from two Nevzorov probes? If yes, what were the differences?

Yes, we had two Nevzorov probes installed for the RadSnowExp project. One is own by NRC and the other owned by ECCC. At this moment, the processed data from the ECCC Nevzorov probe are not available to us.

5. Did you try to calculate X, Ka, and W reflectivities using your scattering data base and compare them to the radar observed values?

Yes, we simulated them and they are plotted in Figure 3, panels d, e, f. The red (blue) lines show the measurements below (above) the airplane while the black line is the simulated reflectivity at the flight level. As you can see, the simulations are in good agreement with the measurements. To make these panels more readable and consistent with the other panels we plot reflectivities above and below the plane as the shading limits. This reflects how we estimated an uncertainty of radar reflectivity reflectivity at the flight level.

6. Please clarify in more details how the "truth" in Fig. 4c was obtained.

We modified the caption of Figure 4 and added the following statement at the beginning of Sect. 3.3: "Finally, the retrieval results are evaluated against the microphysical properties of snow at the flight level determined using the optimal estimation framework (see Sect. 3.2) These validation data serve as an in-situ "truth"."

7. Line 229: what are the uncertainties of estimating Doppler velocity from a moving aircraft?

The following statement was added in the manuscript for clarity: "Note that, the Doppler velocities as well as the radar reflectivity values at the flight level are not directly measured. They are estimated from the radar measurements below and above the airplane and in-situ probe data using the data assimilation technique that exploits the radar simulator described in Sect. 3.2. Their uncertainties are estimated by propagating the errors on the state vector, $x$"

Editorial comments:

1.  Line 10 and elsewhere: provide units for Dm and IWC to better understand RMSE values here and statistical metrics results given in terms of logarithmical values.

The state vector is in logarithmic units therefore the RMSE values are in bels i.e. they show relative errors (0.15 ~ 40% relative error).

2.  Equation (1): provide integration limits.

The integration limits are provided now. We also added this comment:

"Note that the PSD is a positive function only over a limited set of particle sizes so the effective integration limits are finite."

3.  Line 270: If only one frequency (W) has non-Rayleigh scattering it is already dual-frequency not triple-frequency approach.

We modified this sentence accommodating your comment: "This pinpoints at the shortcomings of the dual- and multi-frequency radar-based approaches…"

---

## Author Comment (AC2)

On behalf of the co-authors, I would like to thank for all the comments and list of corrections that helped to improve the quality of our paper. The detailed responses to the raised issues are provided in the attached file. All the answers are written in red font to make them easier to find.

This manuscript describes microphysical retrievals using multi-frequency radar. Retrievals of ice water content, mean mass diameter, and degree of riming are demonstrated. The manuscript shows the difficulty in retrieving the degree of riming, particularly from a single-wavelength. The authors conclude by demonstrating the retrievals from single, multi-frequency and multi-frequency with Doppler velocity in a research flight. The single frequency has little information content in the degree of riming or in volumes with large drops. The multi-frequency retrievals improve upon the single frequency, but still struggle with degree of riming. By adding a Doppler velocity to multi-frequency, the retrievals are in better agreement with the measured quantities.

Overall this manuscript is straightforward and shows interesting results for a difficult problem. I have a few main comments as well as some smaller things I noticed.

- It seems one of the main take-aways the authors would like the reader to have is the improvement to the microphysical retrievals, particularly the degree of riming, when a Doppler velocity is included. However, it was unclear to me how this is used in the retrieval and seems to come out of nowhere when Fig. 4 is introduced. Is it from nadir or zenith (or some interpolated mean like reflectivity)? Does the Doppler velocity include air motions? How is it included in the actual retrieval?

To address this issue, we've slightly modified the first paragraph of Sect. 3.3. Now it clearly states what has been used as the radar measurements at the flight level: "The optimally estimated radar measurements and the microphysical data described in the previous subsection are used for the validation of the microphysical retrieval. First, the triple frequency radar reflectivity at the in-situ flight level is used to form the measurement vector (see eq. 7). Then, the expected value of the state vector is estimated using the methodology presented in Sect. 2.3. Finally, the retrieval results are evaluated against the microphysical properties of snow determined using the optimal estimation framework (see Sect. 3.2). These validation data serve as an in-situ "truth". An analogous analysis is repeated for two other retrievals: one that is based on single frequency X-band radar reflectivity only and another one based on triple frequency reflectivity data with the addition of the mean Doppler velocity at the X-band. Note that, the Doppler velocities as well as the radar reflectivity values at the flight level are not directly measured. They are estimated from the radar measurements below and above the airplane and in-situ probe data using the data assimilation technique that exploits the radar simulator described in Sect. 2.2. Their uncertainties are estimated by propagating the errors on the state vector, x (eq. 6)."

- Figure 2 is important but I had a hard time interpreting it. What is the difference between the top row and the bottom row? Please clarify.

The caption of Figure 2 was modified to make it clear. Now it says:
" Panels a, b, c: expected values of $\log_{10}D_m$, $\log_{10}IWC$, $\log_{10}\alpha$ in the $DWR_{X-Ka}$–$DWR_{Ka-W}$ space for $Z_X$= 20 dBZ. Panels d, e, f: uncertainties of the quantities presented in the top row (see the color bar captions). The inverse model is derived from the in-situ airborne PSD measurements during MC3E, IPXEx, OLYMPEX and HAIC/HIWC campaigns."

- Figure 3: I'm confused about how the uncertainties are presented in this figure. If I am interpreting this correctly the retrieval of degree of riming has very little uncertainty middle of the flight leg (~1950 – 2110 UTC) (i.e. no black bars), and similarly the measured IWC in panel b (very small green bars)?

The panels that show the reflectivity measurements have been modified to make them more consistent with the other measurements. Now all the panels show the state vector prior the optimal estimation retrieval in green and the posterior value in black. The shading represents the uncertainty, again green and black colors correspond to the prior and posterior errors, respectively.

Minor comments:

Ln 98: Remove or expand '(REF)'.

'(REF)' was removed

Ln 145: Which WC is this – IWC or LWC?

WC was changed to IWC.

Section 3: It is odd to have "3.0.1" etc. for the sections—remove the .0. and make 3.1, etc.

It was corrected. Thank you for pointing it out.

Figure 3: I'm curious why the IWC are presented in kg/m3? In the text this is stated in g/m3 (such as the uncertainty in ln 210). Similar comment for all plots of IWC.

The units of the IWC in the figures will be consistent with the units in the text.

Figure 4: panel b is misaligned compared to the other 2 panels.

The panel b will be aligned with the rest in the final version of the manuscript.

Ln 241: Please check the sentence beginning "For negligible DWRs, multi-frequency information is reduced so the difference between the algorithms." This is not a complete sentence.

It was modified as:

"For negligible DWRs, the multi-frequency information is reduced to one frequency and the difference between the algorithms disappears."

Ln 255: "estimates is available" should be "estimates are available".

It was corrected.

---

## Author Comment (AC3)

On behalf of the co-authors, I would like to thank for all the comments and list of corrections that helped to improve the quality of our paper. The detailed responses to the raised issues are provided in the attached file. All the answers are written in red font to make them easier to find.

This manuscript describes a snow microphysical property retrieval algorithm that employs multi-frequency radar simulations.   A novel aspect of this study is that no a priori particle size distribution (PSD) parametrization is used as part of the retrieval scheme.   Instead, direct airborne PSD measurements, combined with state-of-the-art ice scattering models, are used in forward three-frequency radar simulations to retrieve microphysical properties that are then compared to independent and concurrent airborne microphysical observations.   Two key findings that are not entirely unexpected, but still extremely valuable as quantifiable evidence for the community's benefit, are that multi-frequency radar and Doppler velocity measurements are key observables needed to produce optimal snow microphysical property retrievals.

Despite being limited to one case study, this study is an extremely useful addition to the literature as a proof-of-concept study that will provide useful guidance on future sensor development to ultimately produce more accurate snow property retrievals.   The snowfall remote sensing community will benefit from lessons learned in this study.   I find the manuscript written in a succinct and easily understandable fashion, yet provides sufficient analytical heft that conveys valuable results.  I encourage its eventual publication after the following minor comments are considered by the authors.

1. Line 48:   Should a different dielectric factor of liquid water be applied to the W-band radar reflectivity forward model simulations?   This is a very basic methodological question, but causes much consternation among researchers applying or modeling radar simulations.  The fact that the authors state that the 0.93 value is appropriate for "standard temperatures and frequencies below the Ka-band" might cause some confusion as to why this value is not altered for W-band simulations.

The convention we use relies on the fact that the dielectric constant of ice is very similar at all the frequency bands considered in this study. This makes the DWR equal to 0 dB for Rayleigh targets at the cloud top. The same convention was used to convert the power measured by the radars to the radar reflectivity so we follow the same approach.

2. Line 52: Minimizing W-band attenuation complications is another novel aspect of this study.   The authors rightly highlight that W-band attenuation must be considered at longer distances from the radar under many circumstances, but the fact that these simulations are created using microphysical observations allows the authors to simplify the proof-of-concept message in the study.

Thank you for a positive feedback.

3. Figure 1 caption: I suggest adding the explicit year of the Morrison and Grabowski reference to the caption for completeness.

The year was added.

4. Lines 56-59 elicit a general methodological question: over what time span are the binned PSD observations aggregated? I cannot offer a strong opinion of the optimal time sampling needed to produce robust binned PSDs, but it would be good to advertise this value to the community. I presume PSD variability over short time scales is deemed somewhat muted for this stratiform event, but I would still appreciate the authors advertising the time scale used for PSD measurements that are utilised in the forward radar reflectivity model.

The following statement was added in the paper:
"All the PSD measurements are aggregated over 5 seconds. At a typical airplane speed of 150 m/s it is equivalent of approx. 8-minute integration for the ground-based instrument (for unrimed snowfall that sediments at approx. 1.5 m/s). This mitigates a problem of undercatchent of large snowflakes that are the most uncommon in the sampling volumes."

5. Line 60: Is the snowflake size automatically measured by analysis software? If so, do appropriate references exist for this procedure?

The details on the procedures for processing the particle images and getting the size of each particle sampled can be found in
https://usermanual.wiki/Document/MANUAL.598477337/help
It is not a standard academic publication thus we do not cite it in the paper. If you are interested in processing raw image data there is a software package called SODA designed to do that. Please see https://github.com/abansemer/soda2 for more detail.

6. Line 84: Minor typographical error. Change to "density *of* ice particles"

It was corrected.

7. Line 98: It appears as if a reference is missing (REF).

'(REF)' was removed.

8. Lines 98-110: This seems like a reasonable and creative method to deal with the complexity of possible snowflake morphology and atmospheric conditions.

Again, thank you for a positive comment.

9. Figure 2 caption: I suggest explicitly writing that the top row are expected values and the bottom row standard deviations of the quantities.

The caption was modified to accommodate your comment.

10. Figure 3: I suggest enlarging the font contained in various figure legends.   The values are very difficult to read.

The font size of the legend in the figures will be increased for the final version of the paper.

11. Figure 3: Panels d, e, and f show reflectivity observations for each radar frequency.  Might it be better to show DWR values instead since DWR is explicitly shown in Fig. 2?  Or somehow creatively combine DWR with the single frequency values shown?   This is not a mandatory suggestion by any means, but I am left wondering if showing DWR observations might also be beneficial to better connect with meaningful information contained in the observations.

We decided to keep three panels with the reflectivity observations but to make them more readable and consistent with the other panels we show the reflectivity measured above and below the plane as the edges of the green shaded areas. By doing so we indicate that these measurements are used as the uncertainty limits for the estimates of the radar observations at the flight level. In addition, it makes it easier to see that the estimate of the reflectivities is within these limits throughout the flight. We acknowledge that the DWR signal might be useful for the reader to better understand the context so we decided to add another panel where these data are shown.